# Cellular and Molecular Mechanism of Pulmonary Fibrosis Post-COVID-19: Focus on Galectin-1, -3, -8, -9

**DOI:** 10.3390/ijms23158210

**Published:** 2022-07-26

**Authors:** Daniela Oatis, Erika Simon-Repolski, Cornel Balta, Alin Mihu, Gorizio Pieretti, Roberto Alfano, Luisa Peluso, Maria Consiglia Trotta, Michele D’Amico, Anca Hermenean

**Affiliations:** 1Department of Infectious Disease, Faculty of Medicine, Vasile Goldis Western University of Arad, 310414 Arad, Romania; danielaoatis@gmail.com; 2Doctoral School of Biology, Vasile Goldis Western University of Arad, 310414 Arad, Romania; 3Doctoral School of Medicine, Vasile Goldis Western University of Arad, 310414 Arad, Romania; simon-repolski.erika@uvvg.ro; 4Department of Pneumology, Arad Clinical Emergency Hospital, 310031 Arad, Romania; 5“Aurel Ardelean” Institute of Life Sciences, Vasile Goldis Western University of Arad, 310144 Arad, Romania; balta.cornel@uvvg.ro; 6Department of Microbiology, Faculty of Medicine, Vasile Goldis Western University of Arad, 310414 Arad, Romania; mihu.alin@uvvg.ro; 7Department of Plastic Surgery, University of Campania “Luigi Vanvitelli”, 80138 Naples, Italy; gorizio.pieretti@unicampania.it; 8Department of Advanced Medical and Surgical Sciences “DAMSS”, University of Campania “Luigi Vanvitelli”, 80138 Naples, Italy; roberto.alfano@unicampania.it; 9Department of Experimental Medicine, University of Campania “Luigi Vanvitelli”, 80138 Naples, Italy; pelusoluisa@alice.it (L.P.); mariaconsiglia.trotta2@unicampania.it (M.C.T.); michele.damico@unicampania.it (M.D.); 10Department of Histology, Faculty of Medicine, Vasile Goldis Western University of Arad, 310414 Arad, Romania

**Keywords:** COVID-19, pulmonary fibrosis, galectin, myofibroblasts

## Abstract

Pulmonary fibrosis is a consequence of the pathological accumulation of extracellular matrix (ECM), which finally leads to lung scarring. Although the pulmonary fibrogenesis is almost known, the last two years of the COVID-19 pandemic caused by severe acute respiratory syndrome coronavirus 2 (SARS-CoV-2) and its post effects added new particularities which need to be explored. Many questions remain about how pulmonary fibrotic changes occur within the lungs of COVID-19 patients, and whether the changes will persist long term or are capable of resolving. This review brings together existing knowledge on both COVID-19 and pulmonary fibrosis, starting with the main key players in promoting pulmonary fibrosis, such as alveolar and endothelial cells, fibroblasts, lipofibroblasts, and macrophages. Further, we provide an overview of the main molecular mechanisms driving the fibrotic process in connection with Galactin-1, -3, -8, and -9, together with the currently approved and newly proposed clinical therapeutic solutions given for the treatment of fibrosis, based on their inhibition. The work underlines the particular pathways and processes that may be implicated in pulmonary fibrosis pathogenesis post-SARS-CoV-2 viral infection. The recent data suggest that galectin-1, -3, -8, and -9 could become valuable biomarkers for the diagnosis and prognosis of lung fibrosis post-COVID-19 and promising molecular targets for the development of new and original therapeutic tools to treat the disease.

## 1. Introduction

Pulmonary fibrosis is a consequence of the pathological accumulation of extracellular matrix (ECM) and blockade of the pathways responsible for the deactivation of pro-fibrotic cells and for removing the proliferated matrix, which finally leads to lung scarring. This complex process involves activation of the acute/chronic immune mechanisms, driven by neutrophils and macrophages, due to their released cytokines, chemokines; myofibroblast transition of epithelial cells; a procoagulant framing in the lung and the oxidative signaling, supported by the accumulation of reactive oxygen species in the lungs; and replacement of the normal type I alveolar epithelium with hyperplastic type II cells [1]. A key role in pulmonary fibrosis progression is the epithelial–mesenchymal interplay, in which abnormal lung mesenchymal cells are related to loss of alveolar type I cells and accumulation of hyperplastic alveolar type II cells, resulting in the accumulation of damaged ECM and lung abnormal architecture remodeling [1].

Although the mechanism by which pulmonary fibrosis progresses is almost known, the last two years of the COVID-19 pandemic caused by severe acute respiratory syndrome coronavirus 2 (SARS-CoV-2) and its post effects added new particularities which need to be explored. SARS-CoV-2 infection has been demonstrated to induce acute respiratory distress syndrome (ARDS) in an estimated 17.2–31% of COVID-19 cases [2,3]. The pulmonary injury seems to be directly related to viral disruption of alveolar epithelial cells and to a large number of infected and uninfected macrophages. Monocyte-derived macrophages migrate to the lung tissue where they become infected resident macrophages and can produce large amounts of pro-inflammatory cytokines and chemokines, which contribute to local tissue inflammation and a harmful systemic inflammatory response called cytokine storm [4]. The severity of the SARS-CoV-2 infections may drive the healing evolution and therefore it is important to evaluate possible post-infection complications, especially lung fibrosis. In fact, post-COVID-19 pulmonary fibrosis may be defined as the presence of persistent fibrotic pulmonary sequelae observed by tomography during clinical visit post-infection, which can be associated with functional impairment [5]. The risk factors for the pulmonary fibrosis development in COVID-19 may be considered advanced age, comorbidities such as hypertension and diabetes, or prolonged Intensive Care Unit stay and duration of mechanical ventilation, a particular cellular level host response [2,6]. McGroder et al. [7] enrolled 76 COVID patients who required supplemental oxygen or mechanical ventilation (42%) during hospitalization and demonstrated four months post-infection that the severity of initial symptoms, duration of mechanical ventilation, LDH on admission, and leucocyte telomere length are independent risk factors for fibrotic-like radiographic abnormalities. Similarly, Patil et al. [8] associated the impact of 600 patients with lung fibrosis post-COVID-19, evaluated at six weeks post-discharge from the hospital, with the severity of the initial disease and the duration, and comorbidities (diabetes). In the absence of the consistent data/protocol available for diagnostic tests, or a specific anti-fibrotic therapy, clinicians suggest a clinical visit and an imaging test, at 1, 3, 6, and 12 months after discharge for those with moderate/severe pneumonia in the infectious phase of COVID-19 [9]. However, clinicians propose complementary strategies to reduce the risk to develop pulmonary fibrosis, such as inhibition of viral replication, a long-standing inhibition of the inflammatory response, or the administration of anti-fibrotic therapy [10]. The time since the onset of the COVID-19 pandemic is short, and it is difficult to clarify the mid- and long-term consequences of COVID-19, but at least we can learn from previous experience with the SARS-CoV-1 epidemic when 8000 patients were infected and 900 deaths. In about one-third of patients, pneumonia was complicated by ARDS. One of the most informative studies enrolled 97 patients which were evaluated by chest radiography one-year after the infection with SARS; abnormalities on radiography were observed in 28% of patients and were positively correlated with functional impairment, whereas 55 patients with two-year follow-up data still have persistent impairment of lung function [11].

## 2. The Main Key Players in Promoting Pulmonary Fibrosis after SARS-CoV-2 Viral Infection

### 2.1. Lung Epithelial Cells and Epithelial–Mesenchymal Transition

The alveolar epithelium is composed of 85–90% of type I alveolar epithelial cells and 10–15% of type II alveolar epithelial cells [12]. The latter perform a number of essential activities for lung functions, as the production of the surfactant to prevent alveolar collapse, and stabilization of the airway epithelial barrier; they are also involved in the immune-defense processes during lung injury and differentiation into type I alveolar epithelial cells to promote tissue repair [13,14].

After an alveolar injury, in the absence of balanced pathways, the pathogenesis progresses to acute respiratory distress syndrome (ARDS), driven by three phases: exudative, proliferative, and fibrosis [15,16]. The exudative phase activates innate cell-mediated immunity which induced permeabilization of vascular endothelium and alterations of the alveolar epithelium, followed by the recruitment of neutrophils, monocytes, and macrophages, promoting a strong lung inflammation known as a “cytokine storm” [17,18]. The last results support a hypothesis in which type II alveolar epithelial cell dysfunctions or injuries promote the initiation of the idiopathic pulmonary fibrosis, which leads to fibrosis and progressive loss of lung function [19]. In turn, the loss of functional cells can limit the repairability of the damaged alveolus. Furthermore, type II alveolar epithelial cells acquire a profibrotic phenotype and produce paracrine mediators that stimulate and activate fibroblasts [19]. In addition, it was found that transforming growth factor-beta (TGFβ) is mainly produced by the epithelial cells which express integrins needed for activation of the inactive form, while cessation of its signaling within the lung epithelium can alleviate pulmonary fibrosis [20,21,22]. Connective tissue growth factor (CTGF) is stimulated by TGFβ signaling and contributes to increasing secretion and deposition of collagen deposition by fibroblasts [23,24]. Studies on bleomycin-induced lung fibrosis showed an increased expression of CTGF in type II alveolar epithelial cells, whereas its blockade can stop fibrosis [25,26].

Epithelial–mesenchymal transition (EMT) is a promoter of the fibrosis process, in which epithelial cells progressively lose the normal phenotype and upregulate profibrogenic markers such as α-smooth muscle actin (α-SMA), fibroblast-specific protein 1 (FSP1), collagen 1, and fibronectin [27]. Some results evidenced the capacity of the alveolar epithelial cells to trans-differentiate into fibrogenic myofibroblasts [28,29]. After SARS-CoV-2 infection, EMT seems to be promoted by neutrophil extracellular traps (NETs) and the secretion of alveolar macrophage factors, as TGF-β, IL-8, and IL-1β, as was noticed in vitro by exposure of alveolar macrophages and neutrophils to the virus [30].

After lung injury, TGF-β overexpressed by damaged epithelial and endothelial cells, but also by macrophages and fibroblasts, stimulates EMT, leading to a positive-stimulation loop. The “canonical” TGF-β signaling pathway is dependent on the activation of the SMAD transcriptional activators, which are responsible for the induction of EMT in epithelial cells [31]. In fact, “non-canonical” TGF-β signaling which involved the extracellular-signal-regulated kinase (ERK) pathway can modulate the EMT trans-differentiation and promote fibrosis [31]. TGF-β1 induced EMT due to its crosstalk with the canonical Wnt/β-catenin pathway [32]; upon TGF-β stimulation, β-catenin is accumulated into the nucleus and promotes EMT in alveolar epithelial cells [33], induced fibroblast proliferation, and pulmonary ECM deposition. The EMT pathogenesis seems to be linked with alveolar epithelial cells autophagy and driven to fibrosis and other lung pathology [34].

Hedgehog signaling is a key regulator of the epithelial–mesenchymal interactions during tissue repair and fibrosis [35,36]. In normal conditions, lung hedgehog signaling maintains fibroblast quiescence and homeostasis [35]. In pathological conditions, as in idiopathic pulmonary fibrosis, hedgehog signaling is overreactive, demonstrated in bleomycin-induced lung fibrosis, while blocking of the hedgehog epithelial–fibroblast trans-differentiation can alleviate experimental pulmonary fibrosis [37,38,39].

Recent results highlighted the potentially critical role of the cellular endoplasmic reticulum (ER) stress in the activation of the unfolded protein response and the production of dysfunctional epithelial cell phenotype that facilitates fibrotic remodeling of the lung tissue in IPF [40]. Additionally, it was demonstrated that ER stress enhances fibrosis through IRE1a-mediated degradation of miR-150 and XBP-1 splicing [41].

### 2.2. Endothelial Cells and Trans-Differentiation to Myofibroblasts

During fibrosis pathogenesis, endothelial cells trans-differentiate into myofibroblasts in a process named endothelial–mesenchymal transition. During the pathological transformation, endothelial cells decrease the expression of specific endothelial markers such as platelet endothelial cell adhesion molecule (PECAM), vascular endothelial cadherin, and increase the expression of pro-fibrogenic markers, including α-SMA, Col-1, and fibronectin [42]. The over-production and deposition of the collagen in idiopathic pulmonary fibrosis is also due to the trans-differentiation of endothelial cells, not only of alveolar cells into myofibroblasts, and their induction is done similarly by TGF-β, Wnt/β-catenin pathways, and PDGF pathways [43].

### 2.3. Lung Fibroblasts and Trans-Differentiation to Myofibroblasts

The lung resident fibroblasts are activated by the pro-inflammatory cytokines secreted from macrophages and T cells (e.g., IL-6, TNF-α, and TGF-β) and they achieve a spindle-shaped phenotype and start to produce collagen. PDGF signaling modulates fibroblast proliferation and differentiation and also promotes TGF-β release from activated macrophages and epithelial cells, contributing to the self-activating loop of collagen secretion and deposition [44]. Other fibrosis promoters are fibroblast growth factor receptors (FGFR-1, -2) which are also fibroblast activators and they contribute essentially to collagen synthesis and deposition driven by FGF-2 [45]. FGFR-1 and FGFR-2 were found to be highly expressed in lung cells, as epithelial and interstitial cells, endothelial, and myofibroblast-like cells of patients with idiopathic pulmonary fibrosis. The resistance to apoptosis in fibroblasts and myofibroblasts is another contributor to fibrosis. The PI3K/AKT/mTOR activation reduces autophagy in fibroblasts and myofibroblasts [46], while the inhibition of EF2K and p38 MAPK signaling decreases autophagy processes, that in turn reduce lung fibroblast apoptosis [47].

### 2.4. Lung Lipofibroblasts

Pulmonary lipofibroblasts, a special type of interstitial fibroblasts which contain typical lipid droplets, are located nearb type 2 alveolar epithelial cells to transfer triglyceride to these epithelial cells [48,49]. Lipofibroblasts interplay with resident lung mesenchymal cells proposed to represent the mesenchymal niche for type 2 alveolar epithelial stem cells [50]. They are heterogeneous populations that expressed different markers, such as platelet-derived growth factor receptor alpha (Pdgfrα^Pos^), Axin2^Pos^, and fibroblast growth factor 10 (Fgf10^Pos^) [50].

Lipofibroblasts may have an important role in the post-COVID-19 effects, especially in obese or diabetic patients [50]. In response to various stimuli, such as hyperoxia and infection [51], pulmonary lipofibroblasts can trans-differentiate from myofibroblasts and contribute to pulmonary fibrosis [52]. Although there is no evidence regarding the mechanism of how the lipofibroblasts promote pulmonary fibrosis after SARS-CoV-2 infection, a positive correlation was suggested between the number of pulmonary lipofibroblasts and the severity of pulmonary fibrosis [53].

### 2.5. Lung Macrophages

The lungs host two types of macrophage populations, based on their origin: resident alveolar macrophages and monocyte-derived macrophages [27]. They have the ability to polarize from M1, a pro-inflammatory phenotype in M2 “alternatively activated status” which is involved in healing and anti-inflammatory activity [54].

There are several findings examining the roles of macrophages in pulmonary fibrosis regulation, being a major source of TGF-β during fibrogenesis [55]. They also promote fibroblast trans-differentiation and proliferation through the secretion of growth factors, such as FGF, VEGF, PDGF, and insulin-like growth factor 1 (IGF-1) [55,56]. Interestingly, macrophages are able to exacerbate fibrosis by IL-1β and CCL18 synthesis [57], or inhibit it by producing matrix metalloproteinases (MMPs), which degrade ECM [58]. Polarized M2 macrophages are responsible for the inflammatory process inhibition and for the fibrotic promotion, through the secretion of chemokines, MMPs, tissue inhibitor of metalloproteinases (TIMPs), and fibronectin. Moreover, M2 macrophages have the ability to trans-differentiate into fibrocyte-like cells that express collagen [59,60].

Recently, it was demonstrated the association of COVID-19-induced “acute respiratory distress syndrome” (ARDS) with an accumulation of monocyte-derived macrophages with significant transcriptional similarities to profibrotic macrophages in idiopathic pulmonary fibrosis [61]. Moreover, in severe COVID-19 cases, the resident alveolar macrophages were found with defective antigen signatures and were severely depleted, and replaced by inflammatory monocytes and monocyte-derived macrophages [62]. Recent results suggested that pro-fibrotic M2 macrophage markers are responsible for the increasing risk of complications after SARS-CoV-2 infection in obese type 2 diabetes patients [63].

## 3. Galectins Promote Lung Tissue Remodeling and Fibrosis Post-COVID-19

### 3.1. Galectin-1 and -8

Galectin-1 (Gal-1) is a member of the galectin family with a high affinity for β galactose-containing oligosaccharides [64]. Gal-1 is a key player in different biological functions, including growth, cell proliferation, inflammation/immune response, and carcinogenesis [65,66,67]. Recently, the involvement of Gal-1 in the progression of idiopathic pulmonary fibrosis has been demonstrated. In hypoxemic conditions, Gal-1 interplays with focal adhesion kinase-1 (FAK1) in lung epithelial cells and contributes to trans-differentiation of fibroblasts into myofibroblasts [68], whereas its inhibition reduced FAK1 activity and alleviated fibrogenesis progression [69].

COVID-19 pathogenesis is characterized by the adaptative-immune stimulation after viral infection and respiratory dysfunction resulting from pulmonary injury and lung hypoxemia. The cytokines storm and hyper inflammation [70] induced endothelial and alveolar damage, followed after 3 weeks by fibrotic features and clinical characteristic symptoms [27]. In this regard, Gal-1 appears to be involved in the COVID-19 pathogenesis, finding a correlation between its blood level, proinflammatory cytokines, and clinical parameters (chest radiography, dry cough); elevated serum Gal-1 values were correlated with IL-1β, IL6, IL-10, IL-23, and IL-33 [71]. Moreover, the statistical analysis highlighted the increased level of IL-10 and Gal-1, as well as a strong positive correlation between them in stage III of COVID-19, suggesting their dependent immunomodulation [71].

Endothelial dysfunctions play a crucial role in SARS-CoV-2 pathology and have been recently demonstrated to be connected with immune cell recruitments and hyper-inflammation and formation of alveolar thrombi by platelets and fibrin. Interestingly, pro-inflammatory mediators interplay with Gal-1, -3, and -8, which act in a concerted manner through the N- and O-linked glycans located on the S viral protein, and assuming to form a galectin-glycan lattice on the surface of the virus and endothelial cells, generated by the angiotensin-converting (ACE2) receptor, integrin β1, and CD44 [72]. However, it was demonstrated that Gal-1 and -8 induce conformational changes in α_IIb_β3-integrin surface receptors on platelets and lead to fibrinogen binding and platelet activation and aggregation [73,74]. The process has been found to involve Ca^2+^ mobilization, phosphorylation of mitogen-activated protein kinases (MAPKs), Akt, and β3 integrin [75,76]. Based on these findings, Gal-1 and -8 may be considered therapeutic targets against viral infection and endothelial dysfunction in the lung microvasculature, against the severe immuno-thrombosis complication of the disease [77], or against trans-differentiation of lung fibroblasts into myofibroblasts, a crucial step in pulmonary fibrosis progression post-COVID-19 [68].

### 3.2. Galectin-3

Gal-3 is another important β-galactoside-binding lectin, and the most studied in terms of involvement in COVID-19 pathology and a possible therapeutic target for this disease [78,79]. Gal-3 modulates the inflammatory response and tissue repair after lung injury [80,81], and is highly expressed in fibroblasts, endothelial cells, and alveolar macrophages [82,83,84,85]. It is well known that Gal-3 plays a crucial role in SARS-CoV-2 infection, not only by being structurally close to the N-terminal domain of coronaviruses spike protein subunit 1 [86], but also by its ability to bind the ACE receptor, which has a structural affinity to ACE2 receptor [87]. Further, Gal-3 is involved in the immune response, modulating cytokine secretion [88], and leading to a cytokine storm syndrome [78,79]. Moreover, the highest blood levels of Gal-3 were found in the severe cases of COVID-19 [89,90]. The COVID patients with acute respiratory failure and Gal-3 serum levels above 35.3 ng/mL, were found to be more likely to develop severe ARDS, but also markedly at higher risk of intensive care unit admission or death [91]. Similarly, Xu et al. [92] already explained its role as a prognostic factor in ARDS. In this regard, Gal-3 may be considered a prognostic biomarker in COVID-19 disease [91].

Going further, Gal-3 was found to be involved not just in the viral infection via spike protein, and in the macrophage-related hyper inflammation phase and cytokine storm, but also in the COVID-19-related pulmonary fibrosis joined to the alveolar damage, edema, and inflammation [87]. Previously, it was demonstrated that the involvement of Gal-3 in promoting TGF-β1 signaling [83] further induces epithelial–mesenchymal transition, ECM production, and apoptosis of alveolar epithelial cells (AECs) in pulmonary fibrosis, whereas inhibiting TGF-β activity reduces PF [29,93,94,95]. Higher Gal-3 level was noticed in the bronchoalveolar lavage of the patients with pulmonary fibrosis, but lower after receiving corticosteroid therapy, whereas Gal-1 overproduction in U937 monocytes was stimulated by TNF-α and interferon-gamma in a positive loop [96]. Severe COVID-19 was associated with hyper inflammation and supported by the concomitant upregulation of Gal-3, TNF-α, and IL-6 in lobar and bronchial pneumonia [97]. In this regard, Gal-3 seems to have an important role in the immune response and inflammation, prior to the development of pulmonary fibrosis. Data show that IL-4 stimulates a Gal-3 autocrine loop with increased expression and secretion of Gal-3 that binds and cross-links CD98 on macrophages [83], via CD98-mediated PI3K alternative macrophage activation pathway [83,98,99]. Moreover, Gal-3 seems to have the ability to bind and activate TLR4 [100] and subsequently induced lung fibrogenesis, while fibroblast-specific deletion of TLR4 in mice induced a significant reduction in lung fibrosis [101]. The fibrogenesis is also supported by a pro-fibrotic macrophage subtype, which is phenotypically characterized by the expression of TREM2 [102]. In hypoxemic condition, Gal-3 binds and activates TREM2 and triggers lung fibrosis [87,103,104]. Similarly, a role of Gal-3 in the microglial pro-inflammatory response and the ability of Gal-3 to further activate TREM2 [105] was demonstrated, which can prevent macrophage apoptosis, promote survival, and support M2 differentiation [106,107].

Fibrosis can be more promoted by Gal-3 through regulating endothelial–mesenchymal transition [108], a key event in the progression of idiopathic pulmonary fibrosis.

### 3.3. Galectin-9

Gal-9 is another lectin involved in SARS-CoV-2 pathogenesis. Firstly, was found that a baseline of 2.042 pg/mL plasma Gal-9 can differentiate SARS-CoV-2-infected from noninfected patients with 95% specificity, whereas a strong correlation with proinflammatory mediators was noticed [109]. Firstly, Gal-9’s role in viral attachment and entry into alveolar epithelial cells in a dependent manner by enhancing the binding affinity of the viral spike protein to alveolar type 2 cells was shown [110]. Another study suggested the potential role of Gal-9 and the T cell immunoglobulin and mucin domain-containing 3 protein (TIM-3) in T cells during the progression of the disease [111], while high levels of the plasmatic Gal-9 cleaved form could be associated with inflammatory markers that reflect the severity of COVID-19 pneumonia [112].

Overall, the role of galectin-1, -3, -8, -9 in the mechanism regulating myofibroblasts activation in pulmonary fibrosis post-COVID-19 is presented in Figure 1.

## 4. Therapeutic Strategies Based on Galectin Inhibitors

Based on the characteristics of galectins to recognize galactose as well as galactose-containing di- and oligosaccharides, the drug discovery efforts have been focused to develop inhibitors that can be options for fibrosis therapies.

Table 1 presents all the ongoing clinical trials targeting the galectins for the treatment of fibrosis. Interestingly, they all have the inhibition of Galectin 3 as target in two different clinical fibrosis settings, namely non-alcoholic steatohepatitis (NASH) with advance fibrosis [113,114] and idiopathic pulmonary fibrosis (IPS) [115,116]. These Galectin 3 inhibitors are all in phase 1 or phase 2 evaluation and have all been compared with placebo.

Particularly, fibrotic NASH participants (18–75 years) were administered with the Galectin 3 inhibitors GR-MD-02 (8 mg/kg) [113] or GB1211 (10 and 100 mg/kg, orally, twice per day over 12 weeks) [114]. Specifically, GB1211 safety and tolerability profile determination is ongoing by analyzing vital signs, electrocardiogram (ECG), adverse events, and clinical parameters along with pharmacokinetics and pharmacodynamics [114]. Instead, GR-MD-02 is in evaluation for mean changes in liver fibrosis determined from LiverMultiScan (LMS), a multi-parametric Magnetic Resonance Imaging (MRI) technique [113]. Regarding the IPS, two different clinical trials administered the Galectin 3 inhibitor GB0139, previously known as TD139, in 18 to 85 adult patients via dry powder inhaler (DPI) [115,116]. Specifically, TD139 has been tested before in a single ascending dose phase 1 study (0.5–1.5–3–10–20–50 mg, once a day for 14 days) and then in a multiple dose expansion cohort, in order to assess the adverse events until 30 days after the first dose, along with its safety and tolerability profile, pharmacokinetics, and pharmacodynamics [116]. Additionally, the efficacy of 3 mg GB0139 (once a day over 52 weeks) is in ongoing evaluation by assessing the annual rate of Forced Vital Capacity (FVC), the time of first respiratory related first hospitalization, the time to death, and the Respiratory Related Quality of Life [115]. Although for two of these clinical trials testing Galectin 3 inhibitors no results have been reported until now [114,115], neither oral GR-MD-02 nor inhaled TD139 has associated to serious adverse events and they seem to significantly modify primary and secondary outcomes compared to placebo groups [113,116].

## Figures and Tables

**Figure 1 ijms-23-08210-f001:**
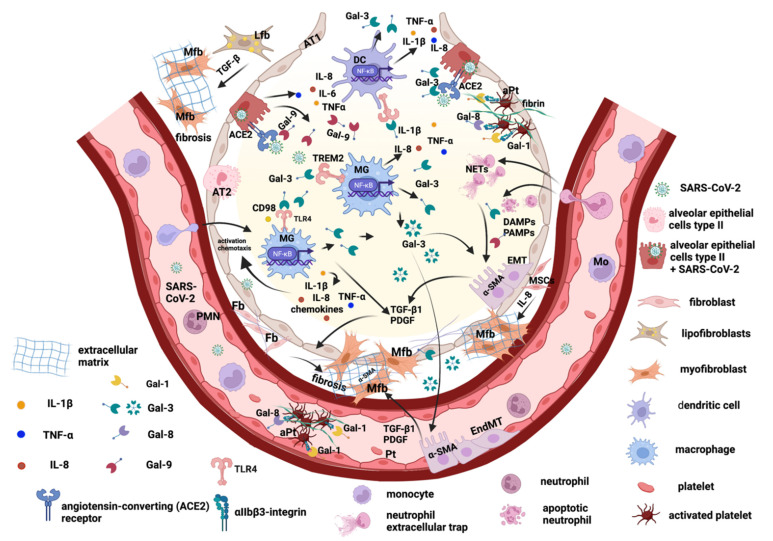
Schematic representation of the galectin-1, -3, -8, and -9 role in the mechanism regulating myofibroblasts activation in pulmonary fibrosis post-COVID-19. macrophages (MG); type I alveolar epithelial cells (AT1); type II alveolar epithelial cells (AT2); dendritic cell (DC); epithelial-mesenchymal transition (EMT); endothelial-mesenchymal transition (EndMT); myofibroblast (Mfb); fibroblast (Fb); lipofibroblasts (Lfb); monocyte (Mo); platelet (Pt); activated platelet (aPt); polymorphonuclear neutrophil (PMN); neutrophil extracellular trap (NETs); angiotensin-converting enzyme 2 receptor (ACE2); mesenchymal stromal cells (MSCs); platelet-derived growth factor (PDGF); transforming growth factor beta1 (TGF-β). This figure was created with BioRender.com (accessed on 18 July 2022).

**Table 1 ijms-23-08210-t001:** Clinical trials targeting the galectins for the treatment of fibrosis.

Sponsor	Compound	Proposed Target	Indication	Phase	NCT Number
Galectin Therapeutics Inc.	GR-MD-02	Galectin 3	Non-alcoholic steatohepatitis (NASH) with advance fibrosis	Phase 2	NCT02421094
Galecto Biotech AB	GB1211	Galectin 3	Non-alcoholic steatohepatitis (NASH) and liver fibrosis	Phase 1b/2a	NCT04607655
Galecto Biotech AB	GB0139	Galectin-3	Idiopathic Pulmonary Fibrosis (IPF)	Phase 2	NCT03832946
Galecto Biotech AB	TD139	Galectin-3	Idiopathic Pulmonary Fibrosis (IPF)	Phase 1/2	NCT02257177

## Data Availability

Not applicable.

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
