# Peer review of "Cellular and Molecular Mechanism of Pulmonary Fibrosis Post-COVID-19: Focus on Galectin-1, -3, -8, -9"

_ijms, 2022, doi:10.3390/ijms23158210_

Round 1

Reviewer 1 Report

The manuscript by Oatis et al proposed to highlight the role of Galectin 1, 3, 8 and 9 on COVID-19 induced pulmonary fibrosis.

The manuscript is interesting and well written. The subject is topical and relevant in the study of the follow-up of post-COVID patients.

I would have some remarks/suggestions to improve the manuscript.

At first, it would be of interest to ameliorate the introduction section, and to mention in the introduction the role of macrophages and other inflammatory cells associated with the cytokine storm observed during COVID infection in the induction of fibrosis

Also, I have trouble understanding the sentence at line 137, could you clarify it?

In the section 2.1, I think it would be of interest to mention the implication/ role of ER stress in pulmonary fibrosis.

I think there is a mistake on line 200. The authors are talking about hypoxia with a reference to hyperoxia, could you clarify?

The authors should go further in paragraph 2.5 which relates to pulmonary macrophages. The paragraph is a bit light in my opinion.

On numerous occasions, the authors discuss the role of hypoxia (line 200, 233, 287, 304), certainly in relation to the hypoxemia observed in COVID patients.

Perhaps it would be appropriate to devote a paragraph to hypoxia in post-COVID induced fibrosis.  

At line 304, the authors discuss the role of TREM2, is there a relationship with M1/M2 differentiation?

Finally, the last paragraph concerning clinical trials is not detailed enough. Are there any publications associated with these clinical trials?

The authors should highlight the clinical trials with the rest of the review.

Author Response

The authors would like to thank this Reviewer for appreciating the novelty of the paper. The answers to the criticism of this Reviewer are below. Please, consider that all changes are highlighted in the Tracked Changes version of the Manuscript.

At first, it would be of interest to ameliorate the introduction section, and to mention in the introduction the role of macrophages and other inflammatory cells associated with the cytokine storm observed during COVID infection in the induction of fibrosis

Response:

we added according to your recommendation.

Also, I have trouble understanding the sentence at line 137, could you clarify it?

 Response:

We changed the phrase

In the section 2.1, I think it would be of interest to mention the implication/ role of ER stress in pulmonary fibrosis.

  Response:

We improved the section 2.1. with ER stress in pulmonary fibrosis data

I think there is a mistake on line 200. The authors are talking about hypoxia with a reference to hyperoxia, could you clarify?

Response: Thank you very much for your observation. We corrected.

The authors should go further in paragraph 2.5 which relates to pulmonary macrophages. The paragraph is a bit light in my opinion.

Response:  We improved the paragraph 2.5.

On numerous occasions, the authors discuss the role of hypoxia (line 200, 233, 287, 304), certainly in relation to the hypoxemia observed in COVID patients. 

Response:  We corrected.

At line 304, the authors discuss the role of TREM2, is there a relationship with M1/M2 differentiation?

 Response: Very valuable your suggestion. We added, accordingly.

Finally, the last paragraph concerning clinical trials is not detailed enough. Are there any publications associated with these clinical trials?

The authors should highlight the clinical trials with the rest of the review.

  Response:

we added according to your recommendation

Reviewer 2 Report

With interest, I read the manuscript ijms-1799573.

In my view, the Authors selected the topic well. More importantly, the addressed galectins in the context of COVID-19 cumulatively, which is an added value.

Overall, the article is well nicely written. However, to the best of my knowledge, the minimal requirement for review article in IJMS is 4,000 words. The present work is more similar to a mini review and thus I encourage the Authors to expand some aspects in the work even more.

In addition, the Authors could increase the number of informative figures in the tables. This would make this review article even more comprehensive. To illustrate what a comprehensive figure/table, one can use current Figure 1.

Finally, a small remark; some details in Figure 1 are difficult to read as too small. Please, make them bigger.

Author Response

Reviewer 2

The authors would like to thank this Reviewer for appreciating the novelty of the paper. The answers to the criticism of this Reviewer are below. Please, consider that all changes are highlighted in the Tracked Changes version of the Manuscript.

Overall, the article is well nicely written. However, to the best of my knowledge, the minimal requirement for review article in IJMS is 4,000 words. The present work is more similar to a mini review and thus I encourage the Authors to expand some aspects in the work even more.

Response: we made changes to the manuscript and we increased substantially the word count.

In addition, the Authors could increase the number of informative figures in the tables. This would make this review article even more comprehensive. To illustrate what a comprehensive figure/table, one can use current Figure 1.

Finally, a small remark; some details in Figure 1 are difficult to read as too small. Please, make them bigger.

Response: We increased the font of the text in Figure 1 and we included a legend, in order to be more “readable”; we kept all the information in a single figure in order to have an overview of the mechanisms driven by all the galectins to promote lung fibrogenesis post-COVID19.

Round 2

Reviewer 2 Report

My comments have been addressed well.